# First eight residues of apolipoprotein A-I mediate the C-terminus control of helical bundle unfolding and its lipidation

Gregory Brubaker[1], Shuhui W. Lorkowski[1], Kailash Gulshan[1], Stanley L. Hazen[1,2], Valentin Gogonea[1,3], Jonathan D. Smith [1,2]*

**1** Department of Cardiovascular and Metabolic Sciences, Cleveland Clinic, Cleveland, Ohio, United States of America, **2** Department of Molecular Medicine, Cleveland Clinic Lerner College of Medicine of Case Western Reserve University, Cleveland, Ohio, United States of America, **3** Department of Chemistry, Cleveland State University, Cleveland, Ohio, United States of America

* smithj4@ccf.org

**Data Availability Statement:** All relevant data are within the paper and its Supporting Information files.

## Abstract

The crystal structure of a C-terminal deletion of apolipoprotein A-I (apoA1) shows a large helical bundle structure in the amino half of the protein, from residues 8 to 115. Using site directed mutagenesis, guanidine or thermal denaturation, cell free liposome clearance, and cellular ABCA1-mediated cholesterol efflux assays, we demonstrate that apoA1 lipidation can occur when the thermodynamic barrier to this bundle unfolding is lowered. The absence of the C-terminus renders the bundle harder to unfold resulting in loss of apoA1 lipidation that can be reversed by point mutations, such as Trp8Ala, and by truncations as short as 8 residues in the amino terminus, both of which facilitate helical bundle unfolding. Locking the bundle via a disulfide bond leads to loss of apoA1 lipidation. We propose a model in which the C-terminus acts on the N-terminus to destabilize this helical bundle. Upon lipid binding to the C-terminus, Trp8 is displaced from its interaction with Phe57, Arg61, Leu64, Val67, Phe71, and Trp72 to destabilize the bundle. However, when the C-terminus is deleted, Trp8 cannot be displaced, the bundle cannot unfold, and apoA1 cannot be lipidated.

## Introduction

Apolipoprotein A-I (apoA1) is the major protein in high density lipoprotein (HDL), and one of the most abundant proteins in human plasma with average levels of ~100 to 150 mg/dl. The assembly of HDL *in vivo* or by cultured cells is absolutely dependent upon the membrane protein ABCA1, which is defective in Tangier disease [1]. However, cell-free reconstituted HDL (rHDL) can be formed from the spontaneous reaction of apoA1 with liposomes made of the short chain phospholipid dimyristoylphosphatidylcholine (DMPC) [2]. This reaction has a maximal rate at the DMPC phase transition temperature of ~24˚C, where the boundary between the fluid liquid crystalline and gel phases creates lower phospholipid packing density that allows the entry of water and weak detergents [3]. The apoA1 protein sequence contains a series of 11 and 22-mer partial repeats, many of which can form a class A amphipathic alpha

**Funding:** This work was supported by grants R01 HL128268 (JDS) and R01 HL128300 (SLH) from the National Institutes of Health (www.nih.gov/). KG was supported by Scientist Development Award SDG25710128 from the American Heart Association (www.heart.org). The funders played no role in this study or manuscript.

**Competing interests:** S.L.H. K.G. S.W.L. and J.D.S. are named as co-inventors on patents held by the Cleveland Clinic relating to cardiovascular diagnostics and therapeutics. S.L.H. reports being eligible to receive royalty payments for inventions or discoveries related to cardiovascular diagnostics or therapeutics from Cleveland Heart Lab, Quest Diagnostics, and P&G. S.L.H. is a paid consultant for P&G; and has received research funds from P&G, Pfizer Inc., and Roche Diagnostics. All other authors no conflict of interest to declare. We have a very loosely related patent entitled "Oxidant Resistant Apolipoprotein A-I and Mimetic Peptides" US Patent Number US 9,522,950 B2. This does not alter our adherence to PLOS ONE policies on sharing data and materials.

**Abbreviations:** ABCA1, ATP binding cassette transporter A1; ANS, 8-Anilino-1-naphthalenesulfonic acid; ApoA1, Apolipoprotein A-I; DMPC, dimyristoylphosphatidylcholine; HDL, high density lipoprotein; MLV, multilamellar vesicles; NEM, N-ethylmaleimide; WMF, wavelength of maximal fluorescence; ΔC, C-terminal deleted; ΔN, N-terminal deleted; ΔN/C, N and C-terminal deleted.

helical structure, with a hydrophobic surface bordered by positively charged Lys and Arg residues, and opposed by negatively charged Asp and Glu residues [4]. Synthetic class A amphipathic helical peptides such as p18A, made without sequence similarity to apoA1, can themselves act as weak detergents, and solubilize DMPC liposomes as well as act as ABCA1-dependent acceptors of cellular lipids; however, at high concentrations, some of these synthetic peptides can strip cells of lipids in an ABCA1-independent manner [5]. ApoA1, even at high concentrations, does not have the promiscuous ability to accept cell lipids in the absence of ABCA1 [5].

Much about apoA1 function and structure has been learned from studying site-specific substitutions and truncations of apoA1, as well as from various structural studies culminating in the crystal structure of the C-terminal deleted apoA1, solved by Mei and Atkinson in 2011 [6]. This crystal structure includes a folded, primarily alpha-helical, bundle extending from residue 8–112. The apoA1 "consensus" model, built from the crystal structure along with chemical crosslinking, and other biophysical and structural data from the past four decades, has labeled the individual helixes: H1 (residues 8–32), H2 (37–45), H3 (54–64), H4 (68–78), H5 (81–115), and H6 (148–179) [7]. It has long been appreciated that the C-terminal truncation of residues 185–243 (called hereafter the ΔC isoform) is dysfunctional in regard to both its DMPC solubilization and ABCA1-dependent lipid acceptor activities [8–11]. This was not surprising, as the C-terminus is the most hydrophobic region of apoA1. However, combining the ΔC truncation with deletion of the N-terminal residues 1–43 (called hereafter the ΔN isoform) to create the doubly deleted ΔN/C isoform, completely rescues apoA1's activity, demonstrating that all that is required for apoA1 function is the central domain [10,12]. Phillips and colleagues proposed that the C–terminus in full length apoA1 interacts with lipid and transmits a structural change allowing the unfolding of apoA1's helical bundle revealing its detergent-like activities [13,14]. This model and a similar one from Atkinson and colleagues [6,15] can explain why the ΔC is defective in lipid binding, which is recovered by the ΔN/C isoform. We previously demonstrated that the free energy required for guanidine unfolding of apoA1 isoforms is ranked ΔC > WT > ΔN/C > ΔN [16]. In the present study we provide new details on the role of the N-terminal 8 residues and Trp8 in stabilizing the helix bundle and new data on the need for the helical bundle to unfold for apoA1's lipidation. We support a model that features the central role of Trp8 in maintaining the helical bundle, whose unfolding is required for apoA1's lipidation.

## Materials and methods

### Generation and purification of recombinant human apoA1 and variants

The bacterial expression vector encoding codon-optimized his-tagged human apoA1 has been previously described [17]. All point mutations and deletions were created using the Quick-Change II Mutagenesis Kit (Thermo Fisher). All mutations were confirmed by DNA sequencing. Expression plasmids were transformed into *E. coli* BL21 dE3 pLysS and protein expression was induced in shaking cultures by overnight incubation with 0.5 mM Isopropyl β-D-1-thiogalactopyranoside at room temperature. The resulting cellular pellet was resuspended in B-PER lysis solution (Thermo Fisher) containing Lysozyme, DNaseI, and a protease inhibitor cocktail. The cellular debris was removed by centrifugation and the supernatant was diluted into PBS containing 3 M guanidine-HCl. The denatured histidine-tagged apoA1 was purified using Ni Sepharose HP resin (Amersham Biosciences) followed by imidazole elution. Fractions containing recombinant apoA1 were extensively dialyzed against PBS and analyzed for purity by SDS-PAGE and Coomassie Blue staining. Only samples with >95% purity were used. When indicated apoA1 was reduced in 10 mM DTT, and reductive methylation was performed with

100 mM N-ethylmaleimide (NEM) at 37˚C for 1 hr under nitrogen gas, followed by dialysis in PBS.

## Non-denaturing gradient gel electrophoresis

5 µg of ApoA1 was mixed with 2X Novex Tris-glycine Native Sample Buffer and run on a pre-cast Novex 4–20% Tris-glycine native gel (ThermoFisher) at 120V for 4 hours. The gel was visualized using GelCode Blue protein stain reagent (ThermoFisher). The migration of a high molecular weight standard (GE Healthcare Life Sciences) containing proteins of known Stokes diameter (nm) is shown for reference.

## Far UV spectral analysis by circular dichroism (CD)

Spectra for all apoA1 samples were collected by CD using a Jasco J810 Spectropolarimeter. The samples were read in a quartz cell with a 0.2 cm path length under a constant nitrogen flush at ambient temperature. Three spectra were collected for each sample from 190–250 nm in continuous scanning mode at a bandwidth of 1 nm with a 0.2 nm data pitch. The spectra were normalized to mean residue ellipticity using 115.5 as the mean residue weight for control apoA1. α-helicity was determined using the molar ellipticity at 222 nm as previously described [18].

## Cholesterol efflux activity

RAW264.7 cells in 24-well plates were labeled by overnight incubation with DMEM containing 1% fetal bovine serum and 0.5 µCi/mL [$^3$H]cholesterol (Perkin Elmer). The next day, the labeling mix was removed and the cells were incubated with DMEM + 0.3 mM 8-Br-cAMP for 16 hours to induce endogenous ABCA1 expression. The cells were washed once with DMEM then 5.0 µg/mL apoA1 in 0.5 mL of DMEM + 0.3 mM 8-Br-cAMP was added to each well for incubation at 37˚C for 4 hrs. The media was removed, briefly centrifuged, and 100 µL was added to a scintillation vial for counting radioactivity to measure cholesterol released to the media. The cells remaining on the plate were extracted using hexane:isopropanol (3:2, v:v) and the radioactivity was counted as a measure of remaining cellular cholesterol. Percent efflux is calculated as % media counts / (media + cell counts).

## Dimyristoyl phosphatidylcholine (DMPC) liposome clearance assay

DMPC (Avanti Polar Lipids) was dissolved in 2:1 (v:v) chloroform:methanol and dried under nitrogen in glass vials. The lipid was resuspended in PBS by vigorous vortexing and multiple freeze/thaws to prepare multilamellar vesicles (MLVs). The turbid DMPC stock solution was diluted to 0.20 mg/mL in PBS, which yielded an absorbance at 325 nm of < 0.5 AU. ApoA1 samples were tested for their ability to clarify the DMPC vesicles using a Gemini EM microplate reader (Molecular Devices) by adding 250 µL of DMPC to 20 µg of apoA1 in PBS (total volume of 300 µL/well). Sample absorbance at 325 nm to assess turbidity was measured over a time course at 24˚C.

## 8-Anilino-1-naphthalenesulfonic acid (ANS) fluorescence assay

ANS (Sigma Aldrich) at a final concentration of 250 µM was mixed with 30 µg/mL of apoA1 samples. Fluorescence spectra were obtained (excitation 395 nm and emission from 425–575 nm) using a Gemini EM microplate reader (Molecular Devices).

## Fluorescence spectroscopy and guanidine or thermal denaturation

For guanidine denaturation, apoA1 samples were prepared at 10 μg/mL in increasing amounts of guanidine hydrochloride (0, 0.25, 0.5, 0.75, 1.0, 1.25, 1.5, 1.75, 2.0, 2.25, 2.5 M) at room temperature. For thermal denaturation 10 μg/mL apoA1 in PBS was equilibrated at the indicated temperature. For both assays, fluorescence spectra were measured using an excitation of 295 nm and emission from 325–375 nm using a Gemini EM microplate reader (Molecular Devices) at the temperature of equilibration. The WMF was determined after data smoothing using GraphPad Prism software. A red shift is observed as the environment of the four apoA1 tryptophans goes from a hydrophobic to an aqueous environment during the unfolding process.

## ApoA1 structural modeling

The ΔC apoA1 crystal structure (Protein Data Bank accession number 3r2p) [6] and the full length apoA1 consensus model (downloaded from http://homepages.uc.edu/~davidswm/structures.html) [7] were visualized and customized using PyMOL 2.1.1 (Incentive Product).

## Statistical analysis

All data show mean ± SD. Cell well and sample replicates were used and assumed to be parametrically distributed, and as our coefficients of variation were small and no *in vivo* measures were ascertained. Multiple columns of data were compared by one-way ANOVA with Dunnett's posttest against one control column as indicated in the figure legends. Statistics were performed using GraphPad Prism software.

# Results

## Central domain of apoA1 is sufficient for apoA1's cell-free and cellular lipidation

N-terminal His-tagged recombinant human apoA1 (rh-apoA1) was purified for the full length wild type (WT) isoform, along with the N-terminal residues 1–43 deleted (ΔN), the C-terminal residues 185–243 deleted (ΔC), and the N- and C-terminal double deleted (ΔN/C) isoforms, the latter of which contains only the central domain of apoA1. Non-denaturing gradient gel electrophoresis demonstrated migration as dimers for all four proteins, with calculated diameters of 7.34, 6,82, 6.71, and 6.38 nm for the WT, ΔN, ΔC, and ΔN/C isoforms, respectively, closely agreeing with the theoretical diameters based on the changes in isoform mass (Fig 1A). We previously demonstrated by following α-helicity using circular dichroism that all four isoforms can be rapidly unfolded by guanidine, and rapidly refolded upon guanidine dilution [16]. We confirmed prior studies that the WT, ΔN, and ΔN/C isoforms were competent to clear DMPC MLVs, albeit with slightly less activity for the ΔN, and ΔN/C isoforms, while the ΔC isoform was incapable of clearing these liposomes (Fig 1B) [10,12]. This same pattern of activity was also observed for ABCA1-mediated cholesterol efflux from RAW264.7 murine macrophages (Fig 1C).

## ApoA1 helical bundle unfolding of different isoforms

The crystal structure of the lipid-free ΔC apoA1 isoform was determined by Mei and Atkinson [6], revealing a large alpha-helical bundle composed of three segments: segment 1, residues 8–40; segment 2, residues 41–67 that are antiparallel to segment 1; and, segment 3, residues 68–115 that are parallel to segment 1. We hypothesized that the apoA1 C-terminal domain can

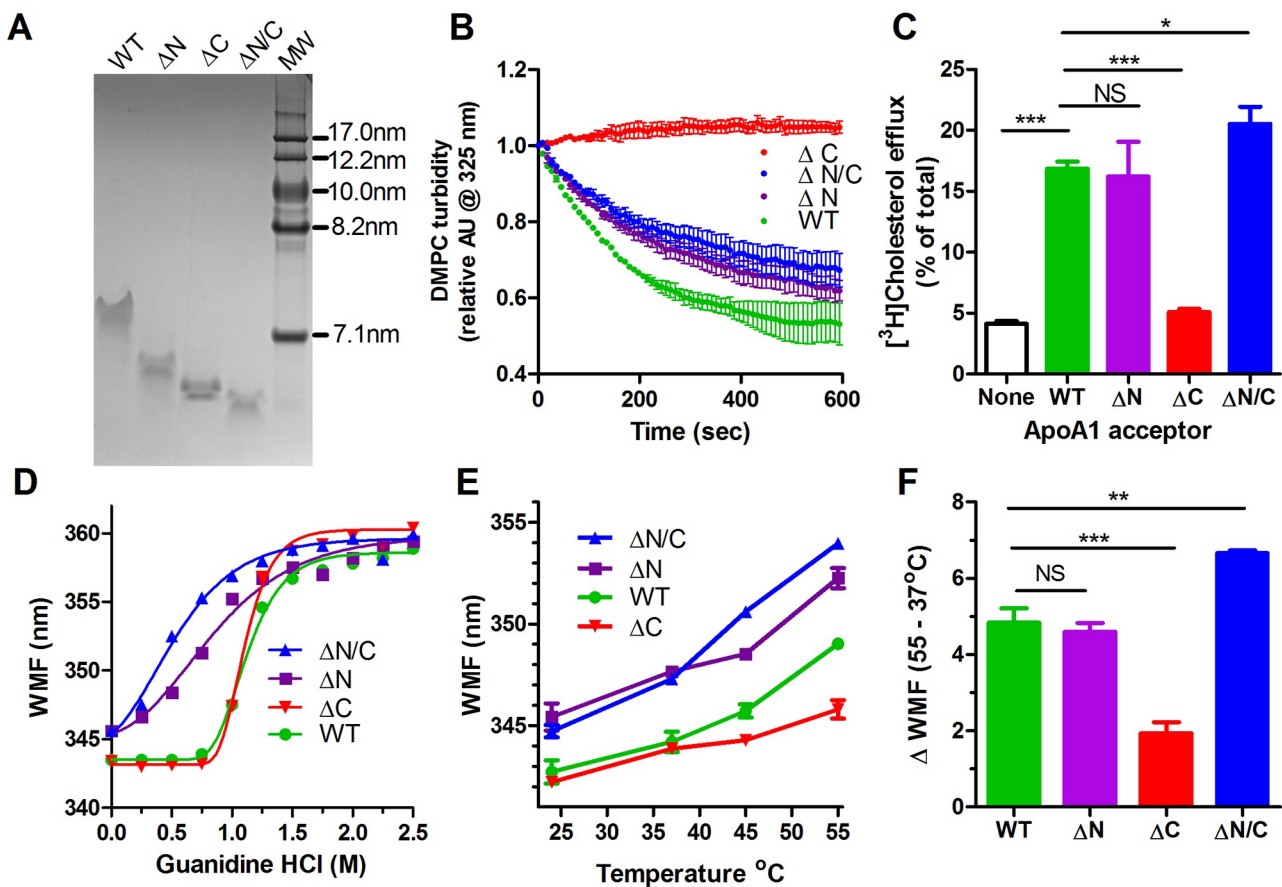

**Fig 1. ApoA1 ΔN/C double deletion partially or fully restores activity of the ΔC isoform. A.** GelCode Blue stained non-denaturing gradient gel electrophoresis of the WT, ΔN, ΔC, and ΔN/C apoA1 isoforms along with a high molecular weight size standard. **B.** DMPC MLV clearance by apoA1. Each line is the average of triplicate wells ± S.D. Different apoA1 isoforms: green, WT; purple, ΔN; red, ΔC; blue, ΔN/C. **C.** Cholesterol efflux from ABCA1-induced RAW264.7 cells to different apoA1 isoforms, as indicated, or in the absence of any acceptor (clear bar black outline). **D.** Guanidine denaturation of apoA1 isoforms assayed by wavelength of maximal fluorescence (WMF) of endogenous Trp residues. **E.** Thermal denaturation of apoA1 isoforms assayed by Trp WMF in triplicate wells ± S.D. **F.** Thermal denaturation of apoA1 isoforms as the delta WMF between 55 and 37°C. The bar graphs are mean ± SD, n = 3 (***, p<0.001; **, p<0.01; *, p<0.05; NS, not significant; by one way ANOVA with Dunnett's posttest compared to WT apoA1).

assist in destabilizing the structure of this bundle and that the unfolding of the bundle is required for apoA1's lipidation. To examine the ability of the various apoA1 isoforms to unfold, we performed guanidine denaturation dose response studies, where unfolding was monitored by the WMF red shift in endogenous Trp fluorescence as it moves from a hydrophobic to hydrophilic environment [9]. There are four Trp residues in apoA1 at positions 8, 50, 72, and 108, all of which are found in the helical bundle, although, Trp8 is just at the boundary. The WMF assay is not dependent upon the concentration of apoA1, and thus is insensitive to small changes in concentration among the four apoA1 isoforms. Examples of the Trp emission spectra for the WT isoform at 0, 1, and 2 M guanidine are shown in S1A Fig. The EC50 for unfolding of the WT and ΔC isoforms was similar at 1.12 and 1.10 M guanidine, respectively (Fig 1D). The ΔN isoform, missing the anchor residues for the segment 1 and the beginning of segment 2 of the helical bundle, unfolded much more readily with an EC50 of 0.82 M guanidine, while the ΔN/C isoform unfolded even more easily with an EC50 of 0.50 M guanidine (Fig 1D). The two N-terminal deletion isoforms (ΔN and ΔN/C) had much lower Hill slopes, indicating less cooperativity in unfolding, vs. the other two isoforms. In addition,

in the absence of guanidine, both N-terminal deletion isoforms have higher basal WMF than the WT and ΔC isoforms. An alternate measure of unfolding using the Trp emission at 345 nm yielded similar results as the WMF plot; although, the unfolding EC50 value was shifted to a lower guanidine concentration for the ΔC isoform (S1B Fig, EC50 = 1.12, 0.76, 1.06, and 0.31 M guanidine for unfolding of the WT, ΔN, ΔC, and ΔN/C apoA1 isoforms, respectively). The % folded by this measure is shown in S1C Fig. We performed circular dichroism on these four isoforms and determined α-helicity of 46.0%, 50.7%, 51.9%, and 50.7% for the WT, ΔN, ΔC, and ΔN/C isoforms, respectively, which are similar to values previously reported for similar recombinant apoA1 isoforms [19,20]. Upon increasing guanidine, we observed decreasing % α-helicity with the ΔN and ΔN/C more sensitive to guanidine, similar to the WMF assay (S1D Fig, EC50 = 1.06, 0.48, 0.97, 0.20 M guanidine for the WT, ΔN, ΔC, and ΔN/C isoforms, respectively) Thermal denaturation was performed in the absence of guanidine and again found that the two N-terminal deletion isoforms (ΔN and ΔN/C) unfolded better (larger WMF shift) than the WT and ΔC isoforms (Fig 1E), which also confirmed the baseline red shift of the two N-terminal deletion isoforms at 24°C. Thermal denaturation also showed that the WT isoform was slightly easier to unfold than the ΔC isoform, thus showing that the deletion of the C-terminus stabilized the folded state (Fig 1F).

## Role of Trp8 and the N-terminal residues in helical bundle unfolding and apoA1 lipidation

The baseline red shift of the N-terminal deletion isoforms could represent either a more unfolded state of the helical bundle or merely the loss of the fluorescence signal from Trp8, which is missing in these isoforms. To evaluate this, we mutated Trp8 to Phe (W8F), Leu (W8L), or Ala (W8A) and we found that the W8F and W8L, two bulky hydrophobic substitutions, retained the baseline WMF of the WT isoform, while smaller W8A substitution increased the baseline WMF (Fig 2A). Subjecting these isoforms to guanidine denaturation showed a stepwise pattern going from least to most sensitive to guanidine unfolding for the aromatic to larger hydrophobic to smaller hydrophobic residues at position 8 (Fig 2A, guanidine EC50: WT, 1.11M; W8F, 0.83M; W8L, 0.77M; W8A, 0.70M).

To further examine the role of Trp8, we made the least stable point mutation W8A, and two new N-terminal deletions 1–7 (Δ1–7) and 1–8 (Δ1–8), all on the ΔC background. The first 7 residues contain 3 prolines in positions 3, 4, and 7 Guanidine denaturation demonstrated that the Δ1–8 and W8A were most sensitive to unfolding, while the Δ1–7 had intermediate sensitivity (Fig 2B). Loss of the hydrophobic C-terminus in the ΔC isoform led to decreased hydrophobicity as ascertained by ANS fluorescence intensity (Fig 2C). The W8A, Δ1–7, and Δ1–8 isoforms on the ΔC background restored its hydrophobicity, indicating increased exposure of the helical bundle (Fig 2C). The unfolding sensitivity completely aligned with the DMPC clearance and ABCA1-mediated acceptor activities of these ΔC isoforms, such that the easiest to unfold isoforms (W8A and Δ1–8) completely rescued the ΔC loss of function, while the Δ1–7 only partially rescued these activities (Fig 2D and 2E). Thus, Trp8 has an essential role in stabilizing the helical bundle, as the least conservative substitution, W8A, or deletion of the first 8 residues, destabilize the bundle and allow for functional recovery of the ΔC isoform.

## Locking apoA1's helical bundle diminished its cell-free and cellular lipidation

To test the role of unfolding of the helical bundle on apoA1's lipidation, we used site directed mutagenesis to replace residues Leu38 and Met112, which in the crystal structure [6] are predicted to be only 3.4 angstroms apart and at the other end of the helical bundle from Trp8,

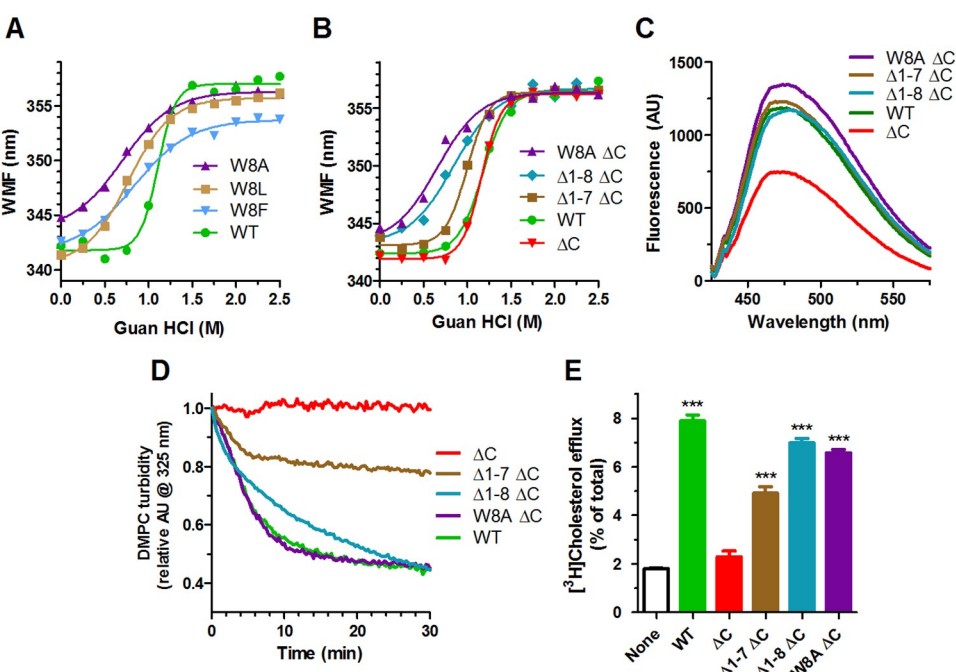

**Fig 2. Role of Trp8 apoA1 helix bundle unfolding and rescue of ΔC isoform activity. A.** Guanidine denaturation of apoA1 isoforms assayed by Trp WMF. Different apoA1 isoforms: green circles, wildtype (WT); blue inverted triangles, W8F; brown squares, W8L; purple triangles, W8A. **B.** Guanidine denaturation of apoA1 isoforms assayed by Trp WMF. Different apoA1 isoforms: green circles, wildtype (WT); red inverted triangles, ΔC; brown squares, Δ1–7 ΔC; blue diamonds, Δ1–8 ΔC; purple triangles, W8A ΔC. **C.** Hydrophobicity of apoA1 isoforms assayed by ANS fluorescence, same colors as in panel B. **D.** DMPC MLV clearance by apoA1, same colors as in panel B **E.** Cholesterol efflux from ABCA1-induced RAW264.7 cells to different apoA1 isoforms or in the absence of any acceptor (clear bar black outline) (average ± SD; ***, p<0.001 by one way ANOVA with Dunnett's posttest compared to no apoA1).

with Cys residues to create the 38C/112C helix bundle locked apoA1 isoform (Fig 3A). After purification, non-reducing SDS PAGE revealed that all of the disulfide bonds are intra-molecular, as we did not observe any dimer sized bands (Fig 3B). The 38C/112C locked apoA1 isoform was harder to unfold in guanidine (EC50 = 1.56 M guanidine) vs. the WT isoform; but, it regained sensitivity to guanidine unfolding upon unlocking the N-hairpin by disulfide reduction with DTT (Fig 3C). The 38C/112C locked apoA1 isoform had very little DMPC MLV clearance activity compared to WT apoA1; however, upon unlocking the helix bundle by reductive methylation with NEM, this activity was largely restored, indicating that loss of this activity was due to the locked helix bundle rather than to the double Cys substitution (Fig 3D). The cellular ABCA1-mediated efflux capacity of the 38C/112C locked apoA1 isoform was reduced by 75% vs. the WT isoform; however, this activity was completely restored by reductive methylation of the disulfide bond using NEM to unlock the helix bundle (Fig 3E). These studies demonstrate that unfolding the helical bundle is necessary in order for apoA1 lipidation via solubilization of DMPC vesicles or via cellular ABCA1 activity.

## Proline substitutions in the helix bundle rescue the activity of ΔC apoA1

To further prove that it is the ability of the helical bundle to unfold that can rescue the activities of the ΔC isoform, we replicated the study of Tanaka et al. [21] and substituted Tyr18 and Ser55, within the N-terminal helix hairpin bundle, with proline residues on the full length and ΔC background isoforms, creating the 2P isoforms. The 2P-WT and 2P-ΔC isoforms are much less folded as they have a large red-shifted baseline WMF, and they are much more sensitive to

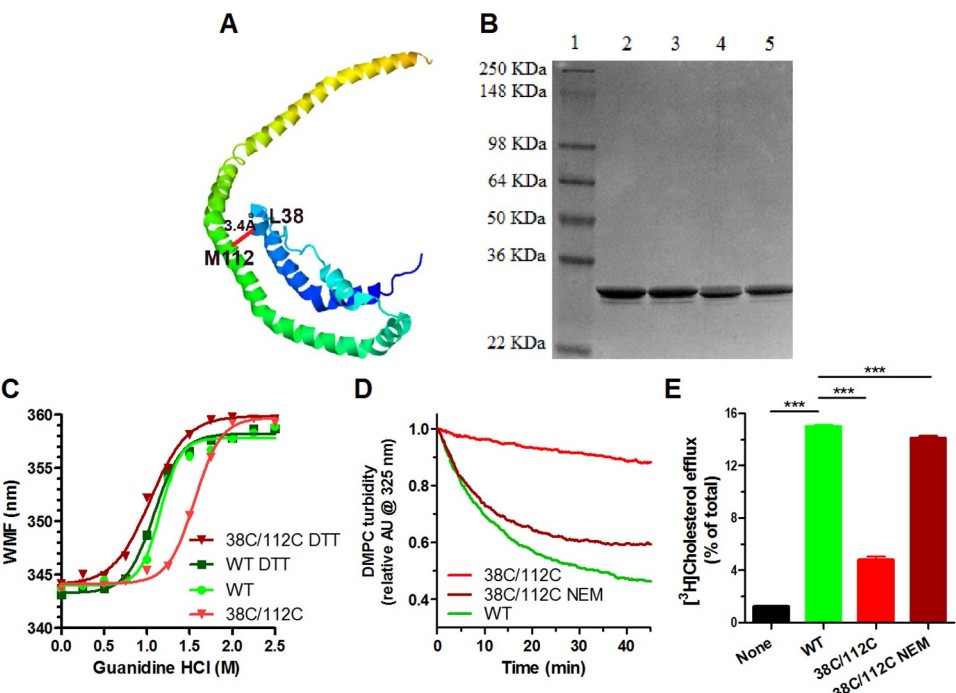

**Fig 3. Disulfide lock of the helical bundle impedes apoA1 activity. A.** Portion of the crystal structure of ΔC apoA1 from (6), showing proximity of L38 and M112 at the "top" of the helical bundle. Color scheme shows N-terminus with dark blue towards the C-terminus in orange. **B.** Coomassie blue stained SDS PAGE of purified recombinant apoA1 isoforms, showing only intramolecular disulfide bonds in the 38C/112C isoform. Lane 1, MW marker; lane 2, WT w/o DTT; lane 3 WT + DTT; lane 4, 38C/112C w/o DTT; lane 5, 38C/112C + DTT. **C.** Guanidine denaturation of apoA1 isoforms assayed by Trp WMF. Different apoA1 isoforms: light green circles, non-reduced WT; dark green squares WT + DTT; inverted bright red triangles, non-reduced 38C/112C; inverted dark red triangles, 38C/112C + DTT. **D.** DMPC MLV clearance by apoA1; green, WT; bright red, non-reduced 38C/112C; dark red, 38C/112C reductively methylated with NEM. **E.** Cholesterol efflux from ABCA1-induced RAW264.7 cells to different apoA1 isoforms, as indicated, or in the absence of acceptor (black bar) (average ± SD; ***, p<0.001 by one way ANOVA with Dunnett's posttest compared to WT apoA1).

guanidine unfolding compared to the WT and ΔC isoforms (Fig 4A). As previously shown [21], the 2P-ΔC isoform completely restored to WT levels the impaired activities of the ΔC isoform in regard to both DMPC clearance and ABCA1-mediated cholesterol efflux (Fig 4B and 4C). Thus, like the W8A, Δ1–8, and ΔN/C isoforms, these specific proline substitutions promote the helical bundle unfolding and rescue the activity deficits of the ΔC isoform, indicating that the C-terminus is dispensable for apoA1 lipidation when the helical bundle is destabilized.

## Discussion

We propose a model for regulation of apoA1's helix bundle unfolding based on the ΔC isoform crystal structure [6], the "consensus" model of apoA1 [7], and our current findings. The crystal structure places Trp8 (W8) at the start of H1 [6], which we propose to be a central player that acts as a "node" to stabilize the helix bundle. W8 sits in a pocket flanked by multiple residues within H3, H4, and the linker region between H3 and H4 (Fig 5). W8 may have hydrophobic interactions with residues F57 (4.1Å distance from W8 in H3), L64 (4.0Å in H3), V67 (4.0 Å in the linker), F71 (5.3 Å in H4) and W72 (3.7 Å in H4). In addition, W8 is reported to have a pi-cation interaction with R61 (3.6 Å in H3) [6], which combined with the above hydrophobic interactions are reported to be the "major forces that hold the helical bundle together [6].

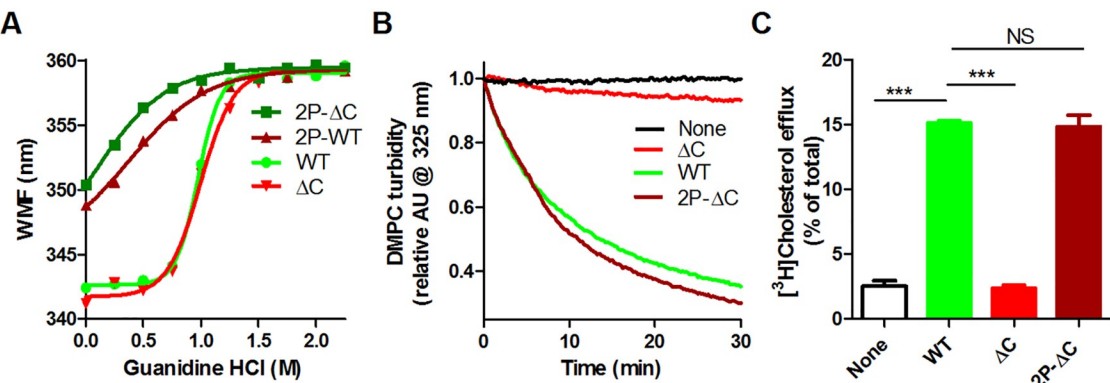

**Fig 4. ApoA1 with two proline substitution in helical bundle restores activity of the ΔC isoform. A.** Guanidine denaturation of apoA1 isoforms assayed by Trp WMF. Different apoA1 isoforms: light green circles, WT; dark green squares 18P/55P (2P-WT); inverted bright red triangles, ΔC; dark red triangles, 2P-ΔC. **B.** DMPC MLV clearance by apoA1 isoforms, or in the absence of apoA1 (black); same colors as in panel A. **C.** Cholesterol efflux from ABCA1-induced RAW264.7 cells to different apoA1 isoforms or in the absence of any acceptor (clear bar black outline), same colors as in panel A (average ± SD; ***, p<0.001 by one way ANOVA with Dunnett's posttest compared to WT apoA1).

Thus, the Trp8 substitution W8A or deletion of residues 1–8 removes the "node" and allows the helical bundle to unfold easier. We speculate that less conservative substitutions of W8 with charged or polar amino acids or with the even smaller hydrophobic Gly residue would also disrupt the stability of apoA1's helical bundle and rescue the dysfunction of the ΔC iso-form similar to the W8A or ΔN isoforms.

In the apoA1 structure consensus model, based on a monomer, the C-terminus is adjacent to the N-terminus (Fig 6A and 6B) [7]. This interaction is supported by 24 distinct lysine/amine intra-molecular crosslinks between the N-terminus/hairpin bundle (residues 1 and 115) and the C-terminus (residues 185 and 243) in full length, monomeric, lipid-free apoA1, as described in four independent studies [7,22–24]. Biophysical studies have previously shown that the hydrophobic C-terminus can undergo a transition to acquire more alpha-helical structure upon lipid binding [25]. We propose that the newly formed amphipathic alpha helical C-

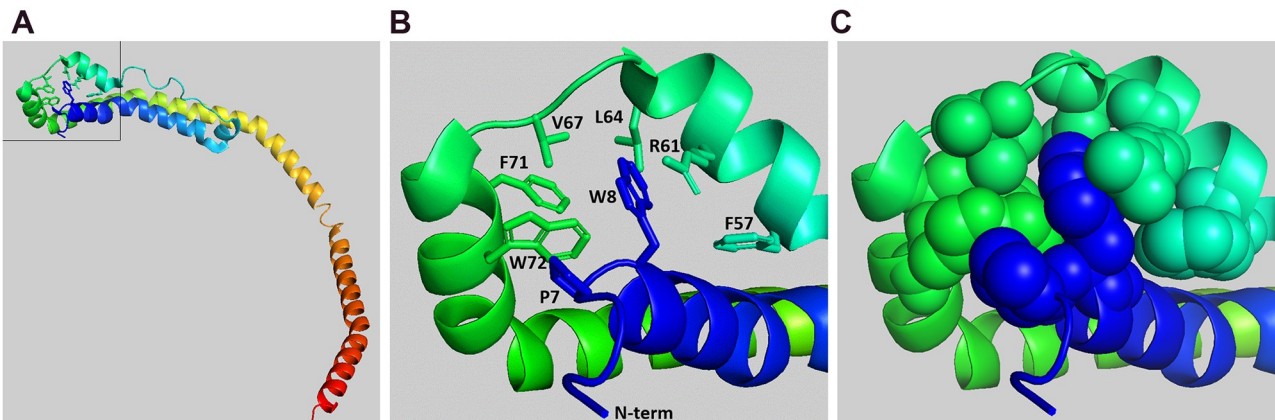

**Fig 5. Trp8 in helix 1 coordinates interaction with residues in helices 2 and 3. A.** Crystal structure of the ΔC isoform from [6] with N-terminus in blue and C-terminus in red. Boxed area is shown in subsequent panels. **B.** Ribbon and stick diagram showing Trp8 and the residues in close proximity in helices 3 and 4 as indicated. **C.** Ribbon and sphere diagram showing Trp8 and the residues indicated in panel B.

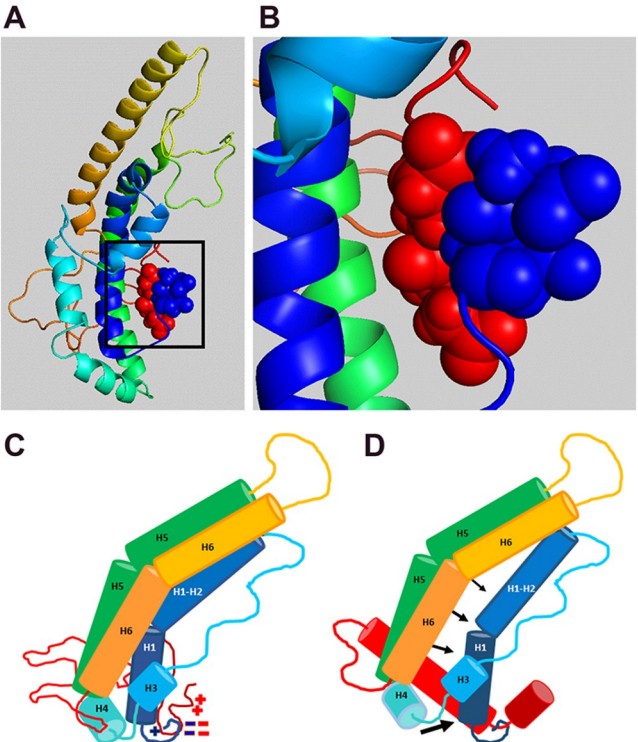

**Fig 6. Model for N- and C-terminal interaction to open helical bundle upon C-terminus lipid sensing. A.** Consensus model of monomeric lipid-free apoA1 from [7] with N-terminus in blue and C-terminus in red, showing close proximity of N-and C-termini. **B.** Blowup of boxed region panel A showing proximity of surfaces for residues D1, E2, E234, and E235. **C.** Cartoon version of ΔC crystal structure from [6] fused to the unstructured C-term region from [7] showing helical segments and proximity of charged residues at N- and C-termini. **D.** Cartoon showing C-terminal helical transformation and extension upon lipid sensing, leading to displacement of the N-terminus. This results in pulling Trp8 out of its position coordinating helices 1 through 4, allowing the unfolding of the helical bundle to expose apoA1's detergent-like amphipathic helices that is required for its lipidation. Helix numbering in C and D is according to the consensus model [7].

terminus tugs on the N-terminal 7-residue appendage to pull W8 out of its pocket (Fig 6C and 6D). This pulling would disrupt W8's interactions with F57, R61, L64, V67, V69, F71, and F72, allowing the helix bundle to unfold so that the newly exposed amphipathic helical hydrophobic surface may bind lipid for HDL assembly. When the C-terminus is deleted, W8 cannot be pulled out from its helix bundle stabilizing position, preventing bundle unfolding and apoA1 lipidation. The bundle destabilizing variants that remove the W8 node (W8A, Δ1–8) at the base of the helix bundle or otherwise destabilize the helical bundle (W18P, S55P in the middle of the helix bundle) can completely rescue the activity of the ΔC isoform. Similarly, the N-terminal 7 residues may also help to stabilize the position of W8, with P7 4.2 Å away from W72, such that the Δ1–7 isoform partially rescues the activity of the ΔC isoform. Our model in Fig 6 is based on the monomeric apoA1 consensus model [7]; however, all of the interactions between the N- and C-termini may in fact be due to intermolecular interactions in dimers. The crystal structure of the ΔC isoform shows two antiparallel dimers, with a hinge region that was modeled to fold back upon itself to form a monomer with some intramolecular surfaces replaced by intermolecular surfaces [6]. All of the studied isoforms are dimers, but our denaturation studies cannot distinguish between inter- and intra-molecular interactions involved in unfolding the hairpin bundle.

Another helical bundle destabilizing isoform (L38G, K40G), in the context of full length apoA1, was also shown to unzip the helical bundle from the opposite end of W8, exposing more hydrophobic surface and leading to faster cellular ABCA1 mediated HDL formation [26]. However, whether this bundle destabilizing isoform could rescue the activity of C-terminal deletion was not determined [26].

Philips and colleagues first proposed that the C-terminal domain senses lipids leading to the opening of apoA1's N-term helix bundle allowing the exposure of its hydrophobic surfaces and its subsequent lipidation [13,14]. This model has been supported by electron paramagnetic resonance evidence showing a structural change in the C-terminal domain upon lipid binding [25], which may drive subsequent unfolding of the helical bundle. Atkinson and colleagues x-ray crystal and apoA1 site directed mutagenesis data support the model that N-terminal helical bundle unfolding is required for lipidation [6,26]. Our current findings extend the evidence for this model by showing that locking the helical bundle with a disulfide bond inhibits apoA1 lipidation, and that W8 plays a central role in coordinating one end of the helical bundle.

All four Trp residues in apoA1 occur in the helical bundle (residues 8, 50, 72, and 108). Our prior *in vitro* mutagenesis work showed that Trp oxidation is responsible for the myeloperoxidase induced loss of apoA1 DMPC solubilization and ABCA1-mediated cholesterol acceptor activities, as the 4WF isoform, with all 4 Trp residues replaced by non-oxidizable Phe residues, is resistant to loss of these activities [27]. Substitution of just Trp72 with Phe (W72F) protects apoA1 from MPO induced loss of activity by ~50%, with the other three Trp to Phe substitutions residues responsible for the other 50% of protection [28]. Thus, the oxidation of the helical bundle Trp residues may either prevent the unfolding of the helical bundle, or alternatively disrupt the amphipathic alpha helical surface required for apoA1 lipidation.

ApoA1 is one of the most abundant plasma proteins with normal levels of ~ 1.5 mg/ml. We hypothesize that apoA1 evolved with its helical bundle structure in order to protect cells from a promiscuous detergent like activity of an extended amphipathic helix. Remaley has previously shown that high concentrations of amphipathic helical peptides can induce lipid efflux from cells even in the absence of ABCA1 expression; while apoA1 induced lipid efflux is dependent upon ABCA1 expression [5]. We demonstrated that ABCA1 mediates the flop of both phosphatidylinositol 4,5 bisphosphate and phosphatidylserine, the former of which is required for apoA1 binding to ABCA1 expressing cells, and the latter of which increases cholesterol extractability to apoA1 or other weak detergents [29]. We further showed that apoA1 binding to ABCA1 expressing cells leads to unfolding of the helical bundle by the use of self-quenching fluorescent probes at positions 38 and 112 [30]. Thus, it appears that apoA1 and ABCA1 co-evolved to regulate apoA1 helical bundle unfolding in order to tightly regulate the detergent-like activity of the abundant plasma protein apoA1.

## Supporting information

**S1 Fig. Guanidine unfolding of apoA1. A.** An example of WT apoA1 Trp fluorescence emission scans at 0 (green), 1.0 (blue), and 2.0 M (purple) guanidine hydrochloride. **B.** Guanidine unfolding of the WT (green), ΔN (purple), ΔC (red), and ΔN/C (blue) apoA1 isoforms assessed using the Trp fluorescence emission at 345 nm. The EC50 = 1.12, 0.76, 1.06, and 0.31 M guanidine for unfolding of the WT, ΔN, ΔC, and ΔN/C apoA1 isoforms, respectively. **C.** The % folded using the single wavelength emission data at 345 nm was calculated by normalizing the data to the 0 M guanidine, and considering full unfolding at 2.0 M guanidine. **D**. Circular dichroism was used to access % α-helicity at increasing guanidine concentrations. The

EC50 = 1.06, 0.48, 0.97, 0.20 M guanidine for the WT, ΔN, ΔC, and ΔN/C isoforms, respectively.
(TIF)

**S1 Data.**
(XLSX)

## Author Contributions

**Conceptualization:** Gregory Brubaker, Stanley L. Hazen, Valentin Gogonea, Jonathan D. Smith.

**Data curation:** Gregory Brubaker, Jonathan D. Smith.

**Formal analysis:** Gregory Brubaker, Valentin Gogonea.

**Funding acquisition:** Stanley L. Hazen, Jonathan D. Smith.

**Investigation:** Gregory Brubaker, Shuhui W. Lorkowski, Kailash Gulshan, Valentin Gogonea, Jonathan D. Smith.

**Methodology:** Gregory Brubaker, Shuhui W. Lorkowski, Valentin Gogonea.

**Project administration:** Jonathan D. Smith.

**Resources:** Jonathan D. Smith.

**Software:** Gregory Brubaker, Valentin Gogonea.

**Supervision:** Jonathan D. Smith.

**Validation:** Gregory Brubaker, Jonathan D. Smith.

**Visualization:** Gregory Brubaker, Valentin Gogonea, Jonathan D. Smith.

**Writing – original draft:** Gregory Brubaker, Jonathan D. Smith.

**Writing – review & editing:** Gregory Brubaker, Shuhui W. Lorkowski, Kailash Gulshan, Stanley L. Hazen, Valentin Gogonea, Jonathan D. Smith.

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
