## [Decision Letter · Decision Letter 0]

4 Sep 2019

PONE-D-19-23187

First eight residues of apolipoprotein A-I mediate the C-terminus control of helical bundle unfolding and its lipidation

PLOS ONE

Dear Dr Jonathan D. Smith,

Thank you for submitting your manuscript to PLOS ONE. After careful consideration, we feel that it has merit but does not fully meet PLOS ONE’s publication criteria as it currently stands. Therefore, we invite you to submit a revised version of the manuscript that convincingly addresses the points raised by both the reviewers.

Since some of the points such as those involving unfolding and aggregation issues may require new experiments, we would appreciate receiving your revised manuscript by October 30th. To enhance the reproducibility of your results, we recommend that if applicable you deposit your laboratory protocols in protocols.io, where a protocol can be assigned its own identifier (DOI) such that it can be cited independently in the future. For instructions see: http://journals.plos.org/plosone/s/submission-guidelines#loc-laboratory-protocols

We look forward to receiving your revised manuscript.

Kind regards,

Maria Gasset, Ph.D.

Academic Editor

PLOS ONE

Journal Requirements:

2. We note that you have a patent relating to material pertinent to this article. Please provide an amended statement of Competing Interests to declare this patent (with details including name and number), along with any other relevant declarations relating to employment, consultancy, patents, products in development or modified products etc. Please confirm that this does not alter your adherence to all PLOS ONE policies on sharing data and materials, as detailed online in our guide for authors http://journals.plos.org/plosone/s/competing-interests by including the following statement: "This does not alter our adherence to  PLOS ONE policies on sharing data and materials.” If there are restrictions on sharing of data and/or materials, please state these. Please note that we cannot proceed with consideration of your article until this information has been declared.

Reviewers' comments:

Reviewer's Responses to Questions

**Comments to the Author**

1. Is the manuscript technically sound, and do the data support the conclusions?

Reviewer #1: Partly

Reviewer #2: Yes

2. Has the statistical analysis been performed appropriately and rigorously? 

Reviewer #1: Yes

Reviewer #2: Yes

3. Have the authors made all data underlying the findings in their manuscript fully available?

Reviewer #1: Yes

Reviewer #2: Yes

4. Is the manuscript presented in an intelligible fashion and written in standard English?

Reviewer #1: Yes

Reviewer #2: Yes

5. Review Comments to the Author

Reviewer #1: Brubaker et al. present a characterization of apoA1 lipidation mechanism via unfolding of the N-terminal α-helices. Although the structural information on apoA1 is rather abundant including available structures of different conformations of the protein - particularly from the Davidson group - the authors provide convincing evidence that the very first N-terminal residues including Trp8 have a functional role solubilizing DMPC lipids or in cholesterol transport. Based on fluorescence measurements, the authors propose a mechanism by which unfolding of an N-terminal helix bundle is required for apoA1 lipidation. The experiments are performed with care and the data analyzed correctly. However, major concerns arise from the experiments shown.

1.- Figure 1 shows that deletion of the C-terminus (ΔC mutant) blocks apoA1 functionality, but the additional deletion of the N-terminus (ΔN/ΔC mutant) recovers lipid binding activity. This evidence goes in line with the model of monomeric, full-length apoA1 structure (Melchior et al NSMB 2017), in which the disordered C-terminus caps the N-terminal helix bundle. Lipid binding to the C-terminal domain of apoA1 induces a conformational switch on this region (Oda et al NSMB 2003) that favors the unfolding of the N-terminal helix bundle. The authors present compelling functional data supporting the hypothesis that unfolding of the N-terminal helical bundle is critical for functional lipidation (especially Figure 1B). Then the authors aim to follow the chemical and thermal unfolding of the different variants of apoA1 by monitoring the change in fluorescence maxima (Figure 1C, D). While full-length and ΔC apoA1 constructs contain 4 Trp residues, ΔN variants contain three Trp residues. More importantly, the structure (for instance, PDB code 3r2p) shows that, precisely, helix 1 clusters helices 2-5 in the bundle. Therefore, removal of helix 1 –assuming that the rest of the bundle remains folded- would expose the remaining Trp residues (Trp50, Trp72, Trp108) to the solvent in monomeric ΔN constructs. In this scenario, it is surprising that the authors observe a change in the fluorescence maxima upon chemical and thermal denaturation. One possible explanation would be that removal of helix 1 promotes oligomerization, in such way that the authors are actually following dissociation of oligomers in ΔN and ΔN/ΔC constructs in Figure 1C, D. To discard this possibility, the authors must show that ΔN, ΔC and ΔN/ΔC constructs are monomeric in the experimental conditions used. Full-length apoA1 is already shown to be monomeric in the SDS-PAGE presented in Figure 3B.

2.- Following protein denaturation by the change in fluorescence maxima is not a quantitative method to determine changes in the populations of folded and unfolded conformers. Different degrees of compaction in unfolded proteins will yield fluorescence maxima at different wavelengths. Authors should monitor the change in fluorescence in a single wavelength to follow protein unfolding.

3.- Proteins were purified in mild denaturing conditions. Proper refolding in the experimental conditions used should be assessed (by Circular Dichroism or comparable techniques). This is particularly important for the ΔN, ΔC and ΔN/ΔC deletion mutants. This point is additionally related to the previous point, since CD would not only be informative on the proper folding of the deletion mutants, but could also be very helpful to characterize protein unfolding upon thermal or chemical denaturation (by monitoring the change in ellipticity in a single wavelength, usually a minima characteristic of α-helical structures).

4.- Figure 1A shows that ΔN and ΔN/ΔC mutants partially recover activity. The authors should state that the mutants “only partially” recover the activity (in Figure legend 1A page#9). Do the authors have any explanation for the observed difference between WT and ΔN and ΔN/ΔC mutants in solubilizing DMPC (only about 50% recovery)?

Minor concerns:

1.- Figure 1A should contain error bars.

2.- Authors state that “the C-terminus stabilized the folded state” based on fluorescence data. However, the ΔC mutant shows much lower change in fluorescence maxima upon thermal denaturation, probably because of the high entropic contribution of the disordered C-terminal region present in WT. Thus, it seems that the C-terminus indeed destabilized the folded state, since in its absence the change in unfolded populations is minor compared to the WT.

3.- Text should be carefully edited: last line in the Introduction reads “who’s” and it should be “whose”. 3 lines before the last in page#5: Remove “of” in “counting radioactivity to measure of cholesterol”… Line 8, page 10, should read “completely” rescued instead of “completed”

4.- Abbreviations should be consistent. The protein is called apoA1 and even apoAI. The C-terminal deletion mutant is termed ΔC and also in page#11 it is called Δ183-243 apoA1.

5.- Labeling of the helices in Figures 6C-D would be appreciated.

Reviewer #2: This study aims to characterize determinants of helical bundle unfolding necessary for lipidation of apolipoprotein A-1 that is mediated by the first eight residues that interact with C-terminus of the protein. The approach uses a Cholesterol Efflux and liposome clearance assay that are now standard to estimate the functionally of the apoA1. In addition to both assays, the authors perform denaturation assay in presence of Guanidium Chloride and Temperature influence. The results suggest that the first eight residues, and specially Trp8, are essential to control the unfolding of the helical bundle necessary for further ApoA1 lipidation. Several published work, spanning the last 20 years, have presented evidence consistent with the current manuscript, but nonetheless, this paper adds a little bit more details and tend to clarify some tenets of apoA1 lipidation and its mechanism.

Taken together the results of the paper are suitable for publication in Plos One. However, the are some points that should be amended or clarified before the manuscript may be accepted. These are detailed in the following:

Major revisions:

- Authors used WMF for unfolding experiments as a red shift is produced when the ApoA1 Trp present in the structure go from a hydrophobic to an aqueous environment during the unfolding process. When Trp are exposed to the hydrophilic environment, an increase of the intensity of the fluorescence is generally observed. The manuscript would gain reproducibility if the authors could provide at least one Trp fluorescence spectrum where the red shift and intensity changes are observed, at least for the Wt and �C and �N isoforms

- In the results corresponding to figure 2, authors mutated Trp8 to Phe, Leu and Ala to discuss the role of the aromatic residue as in the N-terminus unfolding. Trp is an aromatic/amphipathic residue, and it is changed to Phe (aromatic/ hydrophobic) and Leu/Ala (hydrophobic). More disruptive Trp8 changes are needed to clarify the key role of Trp8. Mutations as Trp8 to some polar residue (as Asp and/or Lys and/or Ser) and less bulky residue (Gly) look essential to establish the proper function of the Trp8 into the native structure. Finally, more discussion is needed for mutation Trp8 to Tyr or at least, hypothesize what happens in their model when substitution is performed.

- It is a clear effect that deletion of the first eight residues into �C isoform affects into the unfolding process of the helical bundle. Manuscript would improve if the �1-7 and �1-8 could be performed into the Wt protein and observe the possible effect and comparison wit the isoform �1-8 �C.

Minor revisions:

- Authors proposed in the abstract part “we demonstrate that apoA1 lipidation can occur when the barrier to this bundle unfolding is lowered”. Authors should clarify the term “barrier”, if is from a “thermodynamic or energetic” barrier or another type.

- Introduction lane 2-3: in vivo should be in italics.

- In the introduction, it is stated “the C-terminal truncation of residues 183-243 (called hereafter the �C

- isoform), but authors used a C-terminal truncation of residues of 185-243. Authors should explain why residues 183-184 were not deleted and its possible influence.

- Assays with �C and �N are well-designed and provided information but more discussion is needed. Specifically, authors should discuss why �C is not “more resistant” to unfolding as Wt isoform.

- In Figure 1, it is confusing what black box is. It should be explained, at least, in Figure Legend 1

- Authors mutated Leu 38 and Met 112 to create the 38C/112C helix bundle. Authors should explain better why these residues are chosen as in a quick look of the structure there are more possible candidates.

- In figure 3E, black box should be explained at least in Figure Legend 3

- In figure 4B, red line is not explained or mentioned.

- Page 12, authors mentioned “Thus, like Trp18, D18 and �N/C isoforms….”. It is not clear what D18 is as there are not results refereed to it. Authors should clarify the role of D18 in this context.

- Page 13, authors enlisted a large number of cross links that are not relevant in the discussion part or at least in the main text. It would be more illustrative as supplementary or in a table.

6. PLOS authors have the option to publish the peer review history of their article (what does this mean?). If published, this will include your full peer review and any attached files.

Reviewer #1: No

Reviewer #2: Yes: Manuel Bano-Polo

---

## [Author Response · Author response to Decision Letter 0]

21 Oct 2019

The response to reviewers is found within the attached file. We have addressed every point in our response and we made many changes to the manuscript.

---

## [Decision Letter · Decision Letter 1]

12 Nov 2019

PONE-D-19-23187R1

First eight residues of apolipoprotein A-I mediate the C-terminus control of helical bundle unfolding and its lipidation

PLOS ONE

Dear Dr Jonathan D. Smith,

Thank you for submitting your revised manuscript to PLOS ONE. Both reviewers agree in the manuscript improvement, but to defend the final mechanistic model displayed in Figure 6 the oligomerization control is essential. This control was requested in the first review round and could easily provide it. The study deserves publication in PLoS One but only once this control is provided. Therefore, we invite you to submit a revised version of the manuscript that addresses specifically this point.

We would appreciate receiving your revised manuscript by december 12. To enhance the reproducibility of your results, we recommend that if applicable you deposit your laboratory protocols in protocols.io, where a protocol can be assigned its own identifier (DOI) such that it can be cited independently in the future. For instructions see: http://journals.plos.org/plosone/s/submission-guidelines#loc-laboratory-protocols

We look forward to receiving your revised manuscript.

Kind regards,

Maria Gasset, Ph.D.

Academic Editor

PLOS ONE

Reviewers' comments:

Reviewer's Responses to Questions

**Comments to the Author**

1. If the authors have adequately addressed your comments raised in a previous round of review and you feel that this manuscript is now acceptable for publication, you may indicate that here to bypass the “Comments to the Author” section, enter your conflict of interest statement in the “Confidential to Editor” section, and submit your "Accept" recommendation.

Reviewer #1: (No Response)

Reviewer #2: All comments have been addressed

2. Is the manuscript technically sound, and do the data support the conclusions?

Reviewer #1: Partly

Reviewer #2: Yes

3. Has the statistical analysis been performed appropriately and rigorously? 

Reviewer #1: Yes

Reviewer #2: Yes

4. Have the authors made all data underlying the findings in their manuscript fully available?

Reviewer #1: Yes

Reviewer #2: Yes

5. Is the manuscript presented in an intelligible fashion and written in standard English?

Reviewer #1: Yes

Reviewer #2: Yes

6. Review Comments to the Author

Reviewer #1: The authors have thoroughly reviewed the manuscript and improved it. However, major concerns still remain regarding data interpretation and the reply of the authors is not fully convincing.

This reviewer is concerned about the conformational state and energy of the deletion mutants that the authors used to build their hypothesis. I understand that the goal of the study is not to describe the effect of the deletion mutants on the distribution of populations of ApoA1 conformations. But since the authors finally build a mechanistic model on monomeric ApoA1 (Figure 6), they must show that the effects on stability and unfolding presented in the study account on the same protein species as the WT. In this reviewer’s opinion, they still fail to do so.

All the protein constructs are shown to be dimers. Different values of fluorescence maxima at 0 GdmHCl may reflect that deletion mutants are partially unfolded or they adopt a molten globule state. This reviewer asked the authors to follow unfolding at a single wavelength as it is the only method to quantitatively monitor protein denaturation and determine the fraction of folded and unfolded protein at each GdmHCl concentration. This reviewer is unable to find the solicited analysis in the author’s reply. The dependence on the concentration argued by the authors is true, but the authors claim they are using the same protein concentrations in all the fluorescence experiments.

Regarding CD measurements, it is true that quantification of secondary structure based on changes in ellipticity may not be very robust. Authors claim that CD follows secondary structure and the study aims to describe changes in tertiary structure. This reviewer is concerned about the effect that quaternary structure may induce in the fluorescence data since it has been completely neglected from the interpretation. CD would be ideal to show if the deletion mutants adopt a molten globule conformation at 0 M GdmHCl. Thermal denaturing (particularly DSC) would inform on the possible sequential dimer dissociation-unfolding. The authors added a new line showing the values of secondary structure elements. They must show the CD spectra at increasing concentrations of GdmHCl.

All in all, it is still not clear whether the destabilizing effects on the helix bundle upon lipidation of the C-terminal tail happens intramolecularly or intermolecularly. The data presented by the authors is insufficient to answer this question. The authors still should reply or propose an explanation why mutants only partially recover the ability to solubilize DMPC (New Figure 1B).

This reviewer suggests further editing of the text; some minor mistakes are still present.

Reviewer #2: (No Response)

7. PLOS authors have the option to publish the peer review history of their article (what does this mean?). If published, this will include your full peer review and any attached files.

Reviewer #1: No

Reviewer #2: Yes: Manuel Bano-Polo

---

## [Author Response · Author response to Decision Letter 1]

11 Dec 2019

PONE-D-19-23187R1

First eight residues of apolipoprotein A-I mediate the C-terminus control of helical bundle unfolding and its lipidation

Response to Reviewers: 

We thank the editor and the reviewers for their comments, and we have performed additional studies and analysis to address their concerns.

Editor: 

Both reviewers agree in the manuscript improvement, but to defend the final mechanistic model displayed in Figure 6 the oligomerization control is essential. This control was requested in the first review round and could easily provide it. The study deserves publication in PLoS One but only once this control is provided. Therefore, we invite you to submit a revised version of the manuscript that addresses specifically this point.

Response: This is an excellent point, reflecting the last substantive comment of reviewer 1. We copy here our response to Reviewer 1 for this point. 

We showed in Fig 1A that all of our recombinant apoA1 isoforms are dimers in solution. Thus the effects of different apoA1 isoforms on DMPC clearance and cholesterol efflux represent effects on dimers. The guanidine and heat denaturation studies may reflect, in part, dimer dissociation or N-and C-terminal inter-molecular interaction. As the reviewer states, we cannot distinguish intramolecular vs. intermolecular effects. Our model shown in Fig. 6 is based upon the previously published consensus model, which was only constructed for an apoA1 monomer. We realize that the N- and C-terminal interaction shown in Fig 6 could be due to intermolecular interactions, rather than the intramolecular interaction as shown. Thus, we have modified the discussion section as follows: “Our model in Fig 6 is based on the monomeric apoA1 consensus model [7]; however, all of the interactions between the N- and C-termini may in fact be due to intermolecular interactions in dimers. The crystal structure of the �C isoform shows two antiparallel dimers, with a hinge region that was modeled to fold back upon itself to form a monomer with some intramolecular surfaces replaced by intermolecular surfaces [6]. All of the studied isoforms are dimers, but our denaturation studies cannot distinguish between inter- and intra-molecular interactions involved in unfolding the hairpin bundle.”

Reviewer #1:

Comment: This reviewer is concerned about the conformational state and energy of the deletion mutants that the authors used to build their hypothesis. I understand that the goal of the study is not to describe the effect of the deletion mutants on the distribution of populations of ApoA1 conformations. But since the authors finally build a mechanistic model on monomeric ApoA1 (Figure 6), they must show that the effects on stability and unfolding presented in the study account on the same protein species as the WT. In this reviewer’s opinion, they still fail to do so. All the protein constructs are shown to be dimers. Different values of fluorescence maxima at 0 GdmHCl may reflect that deletion mutants are partially unfolded or they adopt a molten globule state. This reviewer asked the authors to follow unfolding at a single wavelength as it is the only method to quantitatively monitor protein denaturation and determine the fraction of folded and unfolded protein at each GdmHCl concentration. This reviewer is unable to find the solicited analysis in the author’s reply. The dependence on the concentration argued by the authors is true, but the authors claim they are using the same protein concentrations in all the fluorescence experiments.

Response: We analyzed the WMF data using a single wavelength, and we have added this data as Supplemental Figure S1B. We added to the text” “An alternate measure of unfolding using the Trp emission at 345 nm yielded similar results as the WMF plot; although, the unfolding EC50 values was shifted to a lower guanidine concentration for the �C isoform (Supplemental Fig S1B, EC50 = 1.12, 0.76, 1.06, and 0.31 M guanidine for unfolding of the WT, �N, �C, and �N/C apoA1 isoforms, respectively). The % folded by this measure is shown in Supplemental Fig S1C.” We did not favor this analysis because we are performing the studies at the same �g/ml protein concentration, and since the isoforms have different lengths and different number of Trp residues (4 or 3), the absolute values are somewhat meaningless, although the changes with guanidine reflect what was observed using the WMF measure, which is the standard in the apoA1 unfolding field. We normalized this as % folded in Supplemental Fig S1C.

Comment: Regarding CD measurements, it is true that quantification of secondary structure based on changes in ellipticity may not be very robust. Authors claim that CD follows secondary structure and the study aims to describe changes in tertiary structure. This reviewer is concerned about the effect that quaternary structure may induce in the fluorescence data since it has been completely neglected from the interpretation. CD would be ideal to show if the deletion mutants adopt a molten globule conformation at 0 M GdmHCl. Thermal denaturing (particularly DSC) would inform on the possible sequential dimer dissociation-unfolding. The authors added a new line showing the values of secondary structure elements. They must show the CD spectra at increasing concentrations of GdmHCl.

Response: We performed new CD assessment at various guanidine concentrations and added a Supplemental Figure and the following text: “We performed circular dichroism on these four isoforms and determined �-helicity of 46.0%, 50.7%, 51.9%, and 50.7% for the WT, �N, �C, and �N/C isoforms, respectively, which are similar to values previously reported for similar recombinant apoA1 isoforms [19,20]. Upon increasing guanidine, we observed % �-helicity decreasing with the �N and �N/C more sensitive to guanidine, similar to the WMF assay (Supplemental Figure S1D, EC50 = 1.06, 0.48, 0.97, 0.20 M guanidine for the WT, �N, �C, and �N/C isoforms, respectively).” 

Comment: All in all, it is still not clear whether the destabilizing effects on the helix bundle upon lipidation of the C-terminal tail happens intramolecularly or intermolecularly. The data presented by the authors is insufficient to answer this question. The authors still should reply or propose an explanation why mutants only partially recover the ability to solubilize DMPC (New Figure 1B). 

Response: We showed in Fig 1A that all of our recombinant apoA1 isoforms are dimers in solution. Thus the effects of different apoA1 isoforms on DMPC clearance and cholesterol efflux represent effects on dimers. The guanidine and heat denaturation studies may reflect, in part, dimer dissociation or N-and C-terminal inter-molecular interaction. As the reviewer states, we cannot distinguish intramolecular vs. intermolecular effects. Our model shown in Fig. 6 is based upon the previously published consensus model, which was only constructed for an apoA1 monomer. We realize that the N- and C-terminal interaction shown in Fig 6 could be due to intermolecular interactions, rather than the intramolecular interaction as shown. Thus, we have modified the discussion section as follows: “Our model in Fig 6 is based on the monomeric apoA1 consensus model [7]; however, all of the interactions between the N- and C-termini may in fact be due to intermolecular interactions in dimers. The crystal structure of the �C isoform shows two antiparallel dimers, with a hinge region that was modeled to fold back upon itself to form a monomer with some intramolecular surfaces replaced by intermolecular surfaces [6]. All of the studied isoforms are dimers, but our denaturation studies cannot distinguish between inter- and intra-molecular interactions involved in unfolding the hairpin bundle.”

Response: This reviewer is still concerned about the difference between the WT, �N, and �N/C isoforms in the DMPC clearance assay in Fig 1B. The DMPC clearance assay is subject to variance due to the rapid initial clearance rate, the position of samples in the 96-well plate, and difference in time to read the different apoA1 isoforms. These affect the 0 time read, which is used to normalize all of the subsequent time points. We use this assay more as a qualitative vs. quantitative assay to determine if apoA1 is active. The cholesterol efflux assay is more physiological and shows that the �N and �N/C isoforms behave similar to the WT isoform.

Comment: This reviewer suggests further editing of the text; some minor mistakes are still present.

Response: we reviewed the manuscript again to identify and fix any remaining errors.

---

## [Decision Letter · Decision Letter 2]

31 Dec 2019

First eight residues of apolipoprotein A-I mediate the C-terminus control of helical bundle unfolding and its lipidation

PONE-D-19-23187R2

Dear Dr. Jonathan D. Smith,

We are pleased to inform you that your manuscript has been judged scientifically suitable for publication and will be formally accepted for publication once it complies with all outstanding technical requirements.

With kind regards,

Maria Gasset, Ph.D.

Academic Editor

PLOS ONE

Additional Editor Comments (optional):

Reviewers' comments:

Reviewer's Responses to Questions

**Comments to the Author**

1. If the authors have adequately addressed your comments raised in a previous round of review and you feel that this manuscript is now acceptable for publication, you may indicate that here to bypass the “Comments to the Author” section, enter your conflict of interest statement in the “Confidential to Editor” section, and submit your "Accept" recommendation.

Reviewer #1: All comments have been addressed

2. Is the manuscript technically sound, and do the data support the conclusions?

Reviewer #1: Yes

3. Has the statistical analysis been performed appropriately and rigorously? 

Reviewer #1: Yes

4. Have the authors made all data underlying the findings in their manuscript fully available?

Reviewer #1: Yes

5. Is the manuscript presented in an intelligible fashion and written in standard English?

Reviewer #1: Yes

6. Review Comments to the Author

Reviewer #1: (No Response)

7. PLOS authors have the option to publish the peer review history of their article (what does this mean?). If published, this will include your full peer review and any attached files.

Reviewer #1: No

---

## [Editor Report · Acceptance letter]

2 Jan 2020

PONE-D-19-23187R2 

First eight residues of apolipoprotein A-I mediate the C-terminus control of helical bundle unfolding and its lipidation 

Dear Dr. Smith:

I am pleased to inform you that your manuscript has been deemed suitable for publication in PLOS ONE. Congratulations! Your manuscript is now with our production department. 

With kind regards,

on behalf of

Dr. Maria Gasset 

Academic Editor

PLOS ONE